# *Helicobacter pylori*-Associated Iron Deficiency Anemia in Childhood and Adolescence-Pathogenesis and Clinical Management Strategy

**DOI:** 10.3390/jcm11247351

**Published:** 2022-12-10

**Authors:** Seiichi Kato, Benjamin D. Gold, Ayumu Kato

**Affiliations:** 1Kato Children’s Clinic, Natori 981-1227, Japan; 2Gi Care for Kids, Children’s Center for Digestive Healthcare, LLC, Atlanta, GA 30342, USA; 3Department of General Pediatrics and Gastroenterology, Miyagi Children’s Hospital, Sendai 989-3126, Japan

**Keywords:** child, gastritis, *Helicobacter pylori*, host genetic factor, iron deficiency anemia, iron demand, iron uptake, sports activity, virulence factor

## Abstract

Many epidemiological studies and meta-analyses show that persistent *Helicobacter pylori* infection in the gastric mucosa can lead to iron deficiency or iron deficiency anemia (IDA), particularly in certain populations of children and adolescents. Moreover, it has been demonstrated that *H. pylori* infection can lead to and be closely associated with recurrent and/or refractory iron deficiency and IDA. However, the pathogenesis and specific risk factors leading to this clinical outcome in *H. pylori*-infected children remain poorly understood. In general, most of pediatric patients with *H. pylori*-associated IDA do not show evidence of overt blood loss due to gastrointestinal hemorrhagic lesions. In adult populations, *H. pylori* atrophic gastritis is reported to cause impaired iron absorption due to impaired gastric acid secretion, which, subsequently, results in IDA. However, significant gastric atrophy, and the resultant substantial reduction in gastric acid secretion, has not been shown in *H. pylori*-infected children. Recently, it has been hypothesized that competition between *H. pylori* and humans for iron availability in the upper gastrointestinal tract could lead to IDA. Many genes, including those encoding major outer membrane proteins (OMPs), are known to be involved in iron-uptake mechanisms in *H. pylori*. Recent studies have been published that describe *H. pylori* virulence factors, including specific OMP genes that may be associated with the pathogenesis of IDA. Daily iron demand substantively increases in children as they begin pubertal development starting with the associated growth spurt, and this important physiological mechanism may play a synergistic role for the microorganisms as a host pathogenetic factor of IDA. Like in the most recent pediatric guidelines, a test-and-treat strategy in *H. pylori* infection should be considered, especially for children and adolescents in whom IDA is recurrent or refractory to iron supplementation and other definitive causes have not been identified. This review will focus on providing the evidence that supports a clear biological plausibility for *H. pylori* infection and iron deficiency, as well as IDA.

## 1. Introduction

*Helicobacter pylori* infection is quite common and this organism colonizes an estimated fifty percent of the world’s populations [1]. However, there is a wide difference in *H. pylori* prevalence among different countries, with infection rates in Latin America and Africa upwards of 70–80% of adults compared to infection prevalence of 20–30% of adults in Canada and U.S. [2]. Gastric colonization with *H. pylori* is usually life-long, which then, eventually and inevitably, induces persistent mucosal inflammation. Long-term *H. pylori* infection can cause pre-cancerous pathology, including gastric atrophy and intestinal metaplasia in adulthood, leading to especially intestinal-type gastric cancer, particularly in at-risk populations [3,4]. Since 1994, *H. pylori* has been classified as a class I carcinogen associated with the development of gastric adenocarcinoma by the World Health Organization (WHO), as well as by the International Agency for Research on Cancer [5].

In the pediatric population, *H. pylori* is also associated with the development of gastritis, peptic ulcer disease (duodenal more than gastric ulcers), in rare cases mucosal-associated lymphoid-type tissue lymphoma, and extra-gastrointestinal diseases, including iron deficiency (ID)/iron deficiency anemia (IDA) and idiopathic thrombocytopenic purpura [6,7]. However, the majority of *H. pylori*-infected children remain relatively asymptomatic without any readily apparent clinical diseases. Significantly, atrophy and intestinal metaplasia are rarely found in *H. pylori*-infected children, and gastric cancer is extremely rare [6,7] (Figure 1). The fact that compared to adults, an abundance of *H. pylori*-infected children demonstrate less severe gastritis and the resultant outcome of severe clinical diseases indicates a down-regulation of host immune response in the early natural history of infection [8]. With data that suggests that *H. pylori* is an “old” pathogen with respect to human evolution, this dampening of the host response in the initial or early infection after gastric mucosal colonization makes biological sense in order to facilitate immune evasion and establish persistent infection in a unique biological niche. The knowledge that there is a wide difference in a clinical spectrum of *H. pylori*-associated disease, including *H. pylori*-associated IDA, between both pediatric and adult populations is very important in understanding the complex pathobiology of human *H. pylori* infection.

It is known that *H. pylori* is closely associated with the development of IDA in children. Unlike in gastric cancer, *H. pylori*-associated IDA occurs commonly in children and adolescents [9]. Improvement of the associated IDA by eradication of *H. pylori* appears to be dependent primarily upon pediatric age groups, which might not be generalizable to adult populations [10]. These facts lead to a hypothesis that the pathogenesis of the IDA differs from that of gastric ulcer or cancer caused by long-term infection of *H. pylori* (Figure 1). There is a possibility that the development of *H. pylori*-associated IDA in children might not depend solely upon mucosal injury and pathologies such as gastric atrophy and intestinal metaplasia.

It is thought that the pathogenesis and clinical outcome of *H. pylori*-associated diseases, including IDA, depend upon multiple factors, including but not limited to bacterial virulence and environmental factors, as well as host genetic and acquired factors. Furthermore, it is the interaction between these bacterial and host factors and the modulation or influence by environmental exposures that affects the host microbiome, i.e., synergistic mechanisms that then result in gastro-duodenal mucosal disease outcome. However, the specific mechanisms underlying *H. pylori*-associated IDA remain poorly understood. In the present review, the biologically plausible and possible pathogenesis of *H. pylori*-associated IDA in childhood and adolescence and the clinical aspects, including management are discussed.

## 2. Clinical Aspects of *H. pylori*-Associated IDA

### 2.1. Iron Absorption in Humans

Non-heme iron accounts for >80% of dietary iron in developed countries [11]. Reduction from ferric (Fe^+++^) to the ferrous (Fe^++^) forms of iron is essential for intestinal absorption, and in this process gastric acid and ascorbic acid play important roles [11]. Non-heme iron is absorbed primarily in the proximal small intestine via the divalent metal transporter-1 expressed in the proximal duodenum mucosa [10]. On the other hand, the mechanism of absorption of heme iron remains to be poorly understood. Important regulators of hepcidin produced by hepatocytes, and, therefore, of systemic iron homeostasis, include the following components; plasma iron concentration, body iron stores, infection and inflammation, and erythropoiesis [10].

### 2.2. Clinical Evidence

It is reported that up to ID or IDA occurs in one-fourth of children with *H. pylori* infection [12]. *H. pylori* infection prevalence increases in the pre-teen and adolescent age groups, and IDA is also more prevalent in those age groups, irrespective of cause [6]. In a pediatric study in Alaska [13], ID was highly prevalent among school-aged children, and *H. pylori* infection was independently associated with ID and IDA. On the other hand, several studies showed no causal relationship between IDA and *H. pylori* infection in children [14,15,16]. In a randomized controlled study in Bangladeshi children [17], it was shown that *H. pylori* infection is neither a cause of ID/IDA nor a reason for treatment failure with iron supplementation.

However, many epidemiological and interventional studies have shown an association between *H. pylori* infection and ID/IDA in children [13,18,19]. In an international multi-center pediatric study [20], there was a significant association observed between *H. pylori* infection and low ferritin concentration in Chile and Brazil but not in United Kingdom. We speculate that these differences could be explained by different host factors in the populations infected in Chile and Brazil, compared to the U.K., and/or different environmental exposures in these geographically distinct populations, thereby leading to specific indigenous microbiome differences in these population, leading to ID/IDA in one and not in the other population. Meta-analyses have shown that a risk of ID and IDA, particularly that which is unexplained, is higher in individuals with *H. pylori* infection than in those without the infection [21,22]. *H. pylori* eradication can reduce the prevalence of ID in children [20]. It has been demonstrated that the eradication of *H. pylori* could improve iron status with IDA [23]. In a meta-analysis with 16 randomized controlled trials [24], it was demonstrated that *H. pylori* eradication therapy plus oral iron supplementation significantly increased levels of hemoglobin, serum iron, and ferritin more than iron supplementation alone. In particular, such effect for hematological indices with anti-*H. pylori* treatment was also shown in patients with moderate or severe IDA.

Recent studies have reported that *H. pylori*-associated IDA is frequently refractory to iron-supplementation therapy or shows recurrent episodes of the IDA once supplementation has been discontinued [25,26]. Such refractory or recurrent natures of *H. pylori*-associated IDA can be resolved by eradication of the bacteria [23,25,27]. Eradication of *H. pylori* results in reversal of long-standing IDA [28,29]. Furthermore, successful eradication of *H. pylori* leads to long-term resolution of refractory or recurrent IDA in Japanese teenagers [27]. Taking together the cumulative evidence described above, it is concluded that *H. pylori* causality on childhood IDA has been established. In addition, it is thought that *H. pylori* plays a central role in the pathogenesis of *H. pylori*-associated IDA.

## 3. Pathogenesis

### 3.1. Gastrointestinal Mucosal Lesions and IDA

IDA can be directly caused by blood loss from *H. pylori*-induced mucosal injury and gastroduodenal lesions such as erosions and ulcerations. A previous study showed that hemorrhagic gastritis was consistent and key finding in the Alaska native population who have a long history of refractory ID and in whom *H. pylori* infection is prevalent [30]. Moreover, some of *H. pylori*-infected Alaska native children had peptic ulcers with active bleeding that is clinically severe enough to require endoscopic hemostasis [31,32]. AGA technical review stresses that any gastrointestinal lesion that causes a mucosal defect can bleed enough to lead to both overt and/or occult blood loss and, therefore, cause IDA [10]. This review also mentioned that the clinical spectrum is broad because many different lesions in distinct gastrointestinal sites are capable of bleeding in an occult manner. In *H. pylori*-infected children, however, the most common clinical diagnosis at diagnostic upper endoscopy is chronic gastritis without any hemorrhagic mucosal lesions [32,33]. In a pediatric multi-center study in Japan [33], it is suggested that blood loss from the gastrointestinal tract rarely causes IDA in children with *H. pylori* infection. Children with *H. pylori*-associated IDA often have no endoscopic hemorrhagic lesions and negative results for fecal occult blood tests using ant-hemoglobin antibodies [27,34]. Blood loss from the gastrointestinal mucosa does not appear to be the primary pathogenetic cause of *H. pylori*-associated IDA /ID in Alaska native children. Thus, upper gastrointestinal mucosal lesions do not appear to play a direct or central role for IDA pathogenesis in *H. pylori*-infected children (Figure 2).

In some Western countries, Celiac disease is important as a cause of IDA, irrespective of *H. pylori* infection [20,35]. In developing countries, other non-*H. pylori* infections, in particular gastrointestinal parasitic infection, should be considered as risk factors of ID/IDA in addition to contributing factors of poor iron intake and low dietary iron bioavailability that occur in parasite endemic regions of the world [20].

### 3.2. Impaired Gastric Acid Secretion

Impaired iron absorption due to reduced gastric acidity and ascorbic acid concentration, both of which are related to *H. pylori* gastritis and, in particular, atrophic gastritis, are suggested as important pathogenetic factors of *H. pylori*-associated IDA in adults [36]. In addition, an association between transient hypochlorhydria often observed as a consequence of acute *H. pylori* infection and IDA is suggested [37,38]. However, *H. pylori* chronic gastritis is often not atrophic in the majority of the infected children [39] (Figure 1). In an international study with *H. pylori*-associated IDA, corpus atrophy was observed in only two patients and no intestinal metaplasia in any children studied [21]. In a recent Chinese study [40], chronic atrophic gastritis was observed in only 4.4% of *H. pylori*-infected children and neither marked atrophy nor intestinal metaplasia was detected in any patients. Conversely, it is reported that interleukin (IL)-1β may influence ID/IDA risk in *H. pylori-*infected children [41,42]. IL-1β is one of the earliest and most important pro-inflammatory cytokines, and has been detected in both in vitro and in vivo studies of *H. pylori* infection, and IL-1β is also a powerful inhibitor of gastric acid secretion [43]. However, gastric acid secretion is not universally impaired in children with *H. pylori* chronic gastritis, although the secretion is markedly increased in those with *H. pylori-*associated duodenal ulcers [44]. It can be concluded that impaired gastric acid secretion does not appear to be a primary or direct cause of *H. pylori*-associated IDA in childhood and adolescents.

### 3.3. Hepcidin

Hepcidin is an iron-regulatory hormone and inhibits macrophage iron release and intestinal absorption, leading to hypoferremia [45]. Hepcidin plays a key mediator of hypoferremia observed and associated with inflammation. Studies have demonstrated that IL-6 induces hepcidin expression in hepatic cells [46]. In *H. pylori-*infected children, IDA may be caused by increased serum hepcidin [47] (Figure 2). Anemia of chronic inflammation is mediated, in part, by the stimulation of hepcidin by cytokines [48]. Therefore, it has been suggested that refractory-increased hepcidin levels may be involved in the failure of patients with *H. pylori* infection to respond to iron [48].

### 3.4. Recent Pathogenetic Hypothesis

It has been recently hypothesized that competition between *H. pylori* and humans for iron availability could lead to IDA [9]. Iron is essential for cell growth and maintenance not only in human hosts but also in the principle metabolic processes of a number of bacteria, in particular *H. pylori.* Most strains of *H. pylori* likely perform some degree of iron metabolism necessary for their survival and reproduction in the gastric environment, may not, by themselves harm the health of most infected hosts. If this hypothesis is right, it is speculated that some of *H. pylori* strains are not harmful for the host [49], and the others aggressively steal bioavailable iron from the host, resulting in ID/IDA. It has been reported that *H. pylori* strains from IDA patients show more rapid growth and enhanced uptake of both ferrous and ferric ions compared to those from non-IDA patients [50]. In the other study [34], however, the degree of bacterial growth was not significantly different between the IDA and control strains. Thus, further studies will be needed to determine the specific phenotype(s) of the infecting *H. pylori* strains and whether the patient is then at risk for ID or IDA.

#### 3.4.1. Iron-Uptake Mechanisms in *H. pylori*

Knowledge of iron-uptake mechanisms in *H. pylori* is still limited. Many bacteria secrete siderophores, high-affinity ferric chelators, and take up ferric iron-siderophore complexes via specific outer membrane proteins (OMPs) [51]. Although *H. pylori* does not have an identified siderophore or its specific receptor, the microorganism expresses several proteins associated with iron metabolism, including the ferric uptake regulator (Fur), high-affinity transporters of ferrous iron (FeoB) and ferric dicitrate (FecA), as well as non-heme iron-containing ferritin (Pfr) [52] (Table 1). In addition, several iron-responsive OMPs are suggested to play roles in *H. pylori* heme uptake [53,54]. Under iron-restricted conditions, *fecA* gene and *frpB* encoding iron-regulated OMP were up-regulated [34], while in iron-replete environments, *H. pylori* expresses a single *fecA3* and *frpB4* OMP [52]. Contrarily, *pfr* gene is down-regulated under iron-restricted conditions. Under iron-restricted conditions, the expression of several genes is up-regulated in a Fur-dependent manner [52,55]. *H. pylori* Fur protein is also a versatile regulator involved in many pathways essential for gastric colonization [55].

#### 3.4.2. Bacterial Virulence Factors

A number of studies demonstrated that *H. pylori* colonizes the stomach with multiple or plural strains and that the degree, severity and phenotype of gastric pathology depends on complex interplay between various *H. pylori* virulence genes, host genetics, and environmental factors [56]. In particular, it is not known at what stage during the natural history of the infection a certain *H. pylori* strain genotype predominates the persisting infection, thereby resulting in a specific disease phenotype or outcome. It is also known that *H. pylori* shows the genetic diversity across species [57]. *H. pylori* virulence genes can be mainly categorized into three classes: those related to adhesion and colonization, those related to their virulence and ability to confer gastroduodenal mucosal injury, and others [58]. With regard to the initial stage of the colonization of *H. pylori* in the stomach, OMPs expressed on the bacterial surface are important as virulence factors and can bind to gastric epithelial cells [59]. The *H. pylori* genome has nearly 60 genes encoding the OMPs [60].

##### *H. pylori* Colonization and Adhesins

Establishment of *H. pylori* colonization in the stomach is the initial and important step for the development of the subsequent related diseases (Figure 2). *H. pylori* neutralizes the gastric mucosal local environment using its urease activity, which, thereby, enables this organism to occupy the unique biologic microaerophilic neutral pH niche at the mucosal surface underneath the strong acidic gastric environment. *H. pylori* move freely in the dense mucosal layer with a bundle of 2–6 unipolar flagella. However, these specific virulent determinants are similar across all viable *H. pylori* strains and, thus, are thought to be non-specific factor for IDA developments. On the other hand, it is suggested that genes that regulate flagellar synthesis might be involved in the regulation of other virulence factors such as adhesins [61].

Among OMPs, the members of *H. pylori* outer membrane protein (Hop) group, sialic acid-binding adhesin (SabA), and blood group antigen-binding adhesin (BabA) are some of the most frequently studied OMPs [62]. *H. pylori* attaches to the surface-adherent mucus and directly to the gastric epithelial cells with several adhesins, such as SabA and BabA [62] (Figure 2). Once *H. pylori* colonization on the epithelium is established, the bacteria release toxins, including cytotoxin-associated gene A (CagA) and vacuolating cytotoxin A (VacA), leading to gastric inflammation and injury. Risk factors of *H. pylori*-associated diseases include the presence of the *cag* pathogenicity island (*cag*PAI) encoding Type IV secretion system and CagA, *vacA* genotypes, OMPs, including BabA and SabA, and outer membrane inflammatory protein OipA [63,64].

BabA and SabA of *H. pylori* allow the bacteria to persistently colonize in the stomach via interaction with Lewis (Le) and sialylated Lewis (sLe) antigens on gastric epithelia cells, respectively [62,65,66]. It is thought that SabA adhesin plays a key role for gastric colonization in the stage of persistent infection, whereas BabA is an important adhesin in the early stage of the infection [62]. The sLe^x^ and sLe^a^ antigens are thought to be important for *H. pylori* adherence and colonization in the stomach [56]. Although these antigens are rarely present in normal gastric mucosa, gastric inflammation induced by *H. pylori* promotes up-regulation of sLe^x^ antigens, resulting in enhancement of especially the SabA-mediated attachment to the gastric epithelium [65]. Among sLe antigens, the main target receptor of SabA protein is sLe^x^ [62]. On the other hand, the majority of *H. pylori* strains express at least one type of Le antigens. Thus, *H. pylori* molecular mimicry in the form of Le antigens on the bacterial cell surface provides effective mechanisms enabling *H. pylori* colonization within the gastric mucosa, thereby effectively evading the host immune response [67]. SabA also plays a role in non-opsonic activation of human neutrophils [68].

##### Major Cytotoxins

Pathogen-associated molecular patterns (PAMPs), such as lipopolysaccharide (LPS) and flagella, are also the major pathogenetic factors of virulent colonizing *H. pylori* strains [69]. The cytotoxins, such as CagA, VacA and PAMPs, activate antigen-presenting cells, dendritic cells, and macrophages, resulting in stimulation of the adaptive immune response through the production of cytokines, including IL-12 and IL-23 [70].

CagA along with a type IV secretion system is thought to play an important role in the development of gastric cancer [67]. However, *H. pylori* strains can be divided into two types, those that possesses CagA and those that do not express the protein, therefore, the presence or absence of CagA is not always a key pathogenetic factor for *H. pylori*-associated disorders [67]. In a Slovenian study, the *cagA* genotype in children was associated with the degree of gastric inflammation [71]. However, *H. pylori* strains isolated from Japanese children exclusively carried the *cag*A gene, but there were no associations between this gene and the severity of gastritis or peptic ulcer disease [72]. The adherence of *H. pylori* to the gastric epithelial cells induces the expression of CagA, which is reported to be regulated by the Fur protein [73]. In a Japanese study [34], however, there was no association observed between expression of *cagA* gene and development of childhood IDA.

VacA is also an important cytotoxin as *H. pylori* virulence factors, which promotes bacterial colonization and survival in the host cells [74,75]. VacA is considered a multifunctional toxin inducing host cell damage [21,30]. VacA stimulates the regulatory T cells and promotes differentiation to effector T cells, resulting in persistent colonization of *H. pylori* in the gastric mucosa [76]. It is suggested that VacA escapes host immune defenses by differentially regulating the expression of host genes related to immune evasion [77]. It is reported that co-expression of CagA with OipA, VacA, and BabA plays synergic effects in the outcome of *H. pylori*-induced gastric pathologies and disorders [78]. It has been further suggested that VacA is synergistically involved in the pathogenesis of childhood *H. pylori*-associated IDA [34].

##### IDA-Specific Bacterial Factors

Studies on an association between *H. pylori*-specific genes and IDA are limited. It must be noted that expression profiles of genes related to iron metabolism can dynamically change under both iron-replete and depleted conditions. Comparative proteomic analysis suggests that particular *H. pylori* polymorphisms could promote IDA [79]. On the other hand, it was also reported that variation of *feoB* or *pfr* gene is not implicated in IDA development [13,80]. Besides these two genes, known genes related to iron uptake/regulation such as *fecA* and *fur*, were not also involved in IDA pathogenesis [34].

Neutrophil-activating protein (NAP) stimulates neutrophil adherence to the gastric mucosa and its activation, leading to a release of IL-12 and IL-23 that facilitate the Th-1 immune response [81,82]. It is reported that genetic polymorphisms in the *napA* gene may be associated with the pathogenesis of IDA [83,84] (Figure 2). Recently, a high expression of *sabA* gene in *H. pylori* has been reported to be associated with IDA in childhood and adolescence [34]. Interestingly, as previously mentioned, this study also shows that *vacA* may play a synergic role in the development of IDA. The *sabA* gene is highly divergent and regulated with complex mechanisms [62]. Although the SabA expression is thought to enable rapid response to changing conditions in the stomach by switching the “on” (functional) “off” (non-functional), its mechanism remains to be elucidated [62]. A higher salt concentration induces a higher SabA transcription level [85]. As mentioned above, SabA is an important adhesin and detectable in approximately 40% of *H. pylori* strains [67]. SabA has an ability of activating neutrophils through non-opsonic mechanism resulting in damages of the gastric epithelial cells [70]. Sab A is associated with an increased risk of atrophic gastritis and gastric cancer, although causative mechanisms to explain this disease-related association remain controversial [65]. These facts lead to the recognition that *sabA* gene is an important virulence factor in the development of childhood *H. pylori*-associated diseases, including *H. pylori*-associated IDA.

#### 3.4.3. Host Factors

##### Increased Iron Demands as Acquired Factors

It is well known that children, particularly during the first five years of life, are at risk of IDA, which is mainly associated with an increased iron demand due to their rapid growth [86]. *H. pylori* acquisition mainly occurs in infants and younger children via oral–oral or fecal–oral routes. However, *H. pylori*-associated IDA rarely occurs in the early years of life. A randomized control study reported that *H. pylori* is not a cause for treatment failure of iron supplementation in children 2–5 years of age [17]. Contrarily, *H. pylori*-associated IDA frequently occurs in school-aged children, suggesting that increased iron demand due to growth spurt and/or participation in sports-related activities is also important in the pathogenesis [9,25]. Choe et al. reported that increased daily iron demand is important in *H. pylori*-associated IDA in adolescent female athletes [87]. However, sporting activity itself is reported to have no association to *H. pylori*-associated IDA/ID, although the number of female adolescents studied is small [88]. Although both pre-school and school-aged children have increased daily iron demand, *H. pylori* seems to play some causal role for IDA development in the latter but not in the former. It is strongly suggested that on underlying mechanisms of *H. pylori*-associated IDA, the infection itself is a necessary condition but not the primary etiological agent.

##### Host Genetic Factors

Host immune response gene polymorphism affects the susceptibility to *H. pylori* infection and the outcome of *H. pylori*-related disorders [42]. In one study of twins with *H. pylori* infection [89], it was suggested that host genetic factors are important in the development of ID or IDA. In a recent study [27], one sibling case with recurrent *H. pylori*-associated IDA showed long-term resolution after successful eradication therapy.

Toll-like receptors (TLRs) belong to the large family of pattern recognition receptors (PRRs) and are important in the innate immunity of the host [90]. Among the 10 types of TLRs identified in humans [91], TLR2 and TLR4 on the gastric epithelial and immune cells are both associated with recognition of LPS composing the cell wall of *H. pylori*, acting as a primary defense against *H. pylori* [60,92]. However, it seems that TLRs, including these two molecules can play opposite roles, either promotion or suppression of *H. pylori* infection as innate immunity [90]. TLR5 initially plays some role for the recognition of *H. pylori* flagellin, but this bacterium appears to develop mechanisms to escape such recognition for the persistent infection [90]. Although this genetic variation may be advantageous for some individuals, such variation may be less favorable outcomes for other ones that harbor certain genotypes associated with excessive immune response [93]. The LPS recognition receptor TLR4 has been shown to be associated with a higher risk of gastric cancer [90]. Thus, we believe that the roles of TLRs for *H. pylori*-associated IDA remain to be studied.

Host genetic factors that affect cytokines may determine differences in the susceptibility or risk of infected individuals to specific *H. pylori*-associated diseases [70]. It is reported that various single-nucleotide polymorphisms (SNPs) of cytokines genes, such as tumor necrosis factor (TNF)-α, IL-1β, and IL-10, are associated with the risk of precancerous gastric pathology, including atrophic gastritis and intestinal metaplasia [60], although such an association remains controversial [70]. The allelic distribution of IL-1β is one of the most popular candidate gene studied on *H. pylori* infection [59]. Polymorphism of pro-inflammatory cytokine genes encoding IL-1, IL-8, IL-10, and TNF-α is associated with increased risk of *H. pylori*-related gastric cancer and duodenal ulcer disease [58,94,95]. The expression of IL-1β gene increases in *H. pylori*-infected children with ID [96] (Figure 2). A study in Brazilian children has shown that high gastric levels of IL-1β can be the link between *H. pylori* infection and ID/IDA in childhood [41]. Increased concentration of gastric IL-1β was an independent predictor for low blood concentration of ferritin and hemoglobin. In a case–control study from Taiwan, it was reported that IL-1β gene polymorphism may influence ID risk in *H. pylori-*infected children [42]. Pediatric patients with *H. pylori* gastritis showed significantly lower levels of serum ferritin, prohepcidin, and IL-6 compared to those with *H. pylori*-negative gastritis and the healthy control [97]. In this latter study, however, *H. pylori* eradication therapy revealed no significant difference in serum ferritin, prohepcidin, or IL-6 levels.

TNF-α is involved in persistent colonization with *H. pylori* in the stomach [98]. It has been shown that specific TNF-α genotypes are at risk for duodenal ulcer and gastric cancer, but other TNF-α genotypes have a protective function against cancer development [59]. TNF-α has no association with *H. pylori*-associated ID/IDA in children [41].

## 4. Clinical Management Strategies

Although *H. pylori* infection once established persists almost for the host life span, an overwhelming majority of the infected persons remain clinically asymptomatic and suffer no consequences related to their infection. Despite *H. pylori* infection being the primary causative agent for gastric cancer, the microorganisms can produce detrimental or beneficial effects [49]. In a recent systematic review and meta-analysis [99,100], there is some evidence of an inverse association between atopy/allergic diseases and *H. pylori* infection. On clinical strategies, it is very important to consider that universal eradication of *H. pylori* may cause more harm than good for the host with the infection [49]. Pediatric guidelines recommend against a test-and-treat strategy in asymptomatic children, even if the purpose of the strategy is prophylaxis for of gastric cancer [6,7]. Such a consideration is necessary especially for the management of this infection in children.

The Maastricht V/Florence Consensus Report has recommended *H. pylori* testing and eradication therapy for patients with unexplained IDA [101]. On this recommendation, however, the level of evidence is very low. Such a low evidence level suggests that the data on *H. pylori*-associated IDA is insufficient, especially in the adult population. The American Gastroenterological Association (AGA) technical review identified low-quality evidence supporting *H. pylori* testing for patients with IDA [10]. This review has suggested the causal role of *H. pylori* for IDA in the select population, in particular in children, although the relationship is unclear in the majority of adult men and postmenopausal women [10]. In this issue, it has been indicated that two of three RTCs that met the inclusion criteria were in the pediatric population. AGA clinical practice guidelines suggest noninvasive *H. pylori* testing for patients with IDA without other identifiable etiology for bidirectional endoscopy, followed by treatment if positive [102]. On the other hand, in the Houston consensus conference [103], idiopathic thrombocytopenia was discussed regarding the identification of appropriate patients for *H. pylori* testing but not IDA.

The updated North American and European joint pediatric guidelines suggest *H. pylori* testing in children with refractory IDA in which other causes have been ruled out [6]. On the other hand, these guidelines strongly recommend against diagnostic testing for *H. pylori* infection as part of the initial investigation in children with IDA. Pediatric guidelines from Japan recommend eradication therapy for *H. pylori*-infected children with IDA in whom IDA is recurrent or refractory over iron supplementation therapy [7]. The level of evidence for this recommendation from the Japanese guidelines is strong. In summary, “test-and-treat” strategy in *H. pylori* infection should be considered for children and adolescents with recurrent or refractory IDA of unknown causes.

## 5. Perspectives

It is beyond doubt that increased iron demands play an important pathogenetic part in *H. pylori*-associated IDA in children and adolescents. On the other hand, both bacterial virulence and host genetic factors remain to be investigated. Definitive factors are also still not identified on gastric carcinogenesis by *H. pylori* infection in adults. In any case, as mentioned previously, physicians should keep *H. pylori* infection in mind as an important differential diagnosis for children with recurrent or refractory IDA.

It is indicated that *H. pylori* is an important causative factor in the vicious cycle of malnutrition and growth impairments [38], although the literature has many confounding variables and *H pylori* infection is a marker of low economic status. These authors also suggested that direct competition between *H. pylori* and the host for iron is an important contributor to IDA. In Gambia, *H. pylori* colonization in early infancy predisposes the child for subsequent development of malnutrition and growth faltering, although the effect did not persist into later childhood [104]. Furthermore, decreased growth velocity improved significantly over time once *H. pylori* eradication was successful in Colombian children [105]. However, the relationship between *H. pylori*-associated ID/IDA and growth impairment in the *H. pylori*-infected children, especially in developing countries, remains to be further characterized. Clearly, further well-designed, population-based investigations are needed on this issue.

## Figures and Tables

**Figure 1 jcm-11-07351-f001:**
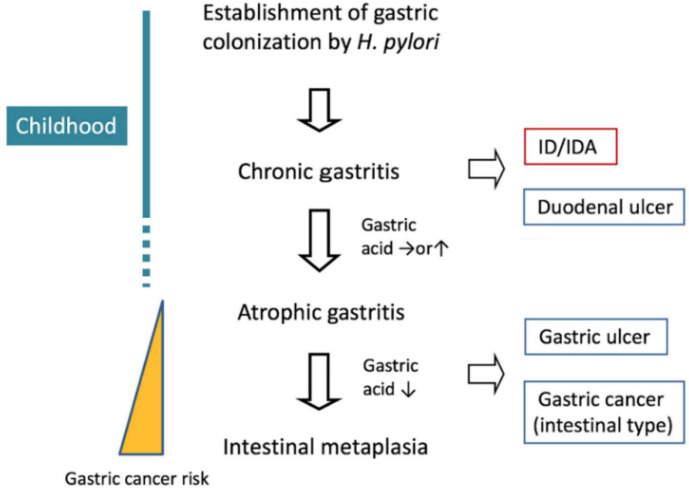
*H. pylori* infection and the related diseases.

**Figure 2 jcm-11-07351-f002:**
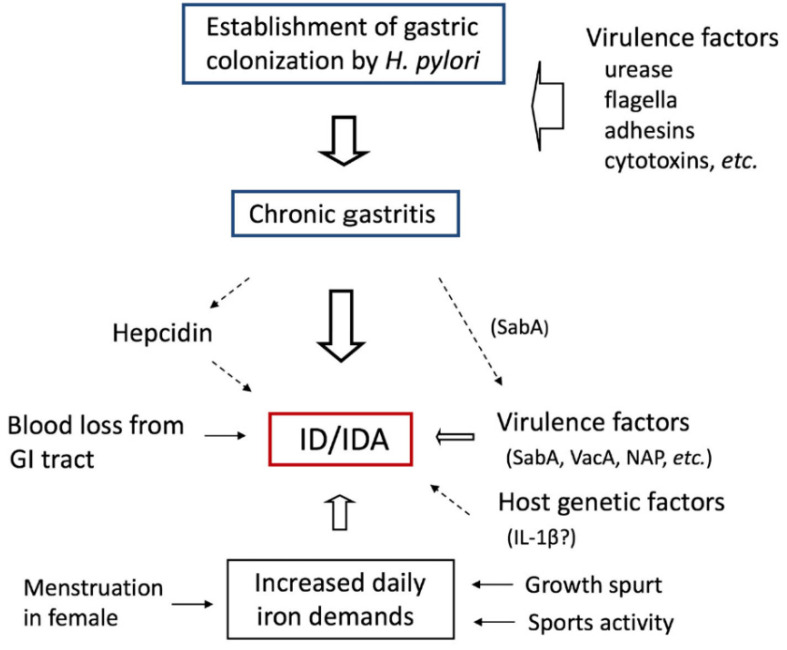
Possible pathogenesis of *H. pylori*-associated IDA in childhood and adolescence.

**Table 1 jcm-11-07351-t001:** Primary genes associated with iron uptake in *H. pylori* host infection and regulation of their expression by iron.

Gene	Function (Hypothetical)	Iron-Deplete Condition	Fur-Regulated
*fur*	Ferric uptake regulator	up-regulation	-
*fecA*	Ferric dicitrate transportor	up-regulation	Yes
*feoB*	Ferrous iron transportor	up-regulation	Yes
*frpB*	Iron-regulated OMP *	up-regulation	Yes
*pfr*	Iron-containing ferritin	down-regulation	Yes
*ceuE*	Iron-transport protein	no regulation?	No

* Outer membrane protein.

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
