# Peer review of "Helicobacter pylori-Associated Iron Deficiency Anemia in Childhood and Adolescence-Pathogenesis and Clinical Management Strategy"

_jcm, 2022, doi:10.3390/jcm11247351_

Round 1
Reviewer 1 Report
This review aimed to demonstrate that H. H. pylori infection, iron insufficiency, and iron deficiency anemia are biologically likely connections. This assessment also tried to educate readers on H. pylori infection management, iron deficiency anemia, and the significance of appropriate clinical follow-up for these individuals. Although the review is well-written and interesting, I believe the abstract might be tighter and should concentrate on the most crucial information.
There are several reviews on this topic such as:
- Zhang Y, Bi J, Wang M, Deng H, Yang W. Correlation between helicobacter pylori infection and iron deficiency in children. Pak J Med Sci. 2022 May-Jun;38(5):1188-1192
- Okuda M, Lin Y, Kikuchi S. Helicobacter pylori Infection in Children and Adolescents. Adv Exp Med Biol. 2019;1149:107-120.
- Sabbagh P, Javanian M, Koppolu V, Vasigala VR, Ebrahimpour S. Helicobacter pylori infection in children: an overview of diagnostic methods. Eur J Clin Microbiol Infect Dis. 2019 Jun;38(6):1035-1045.
- Mehrabani S. Helicobacter pylori Infection in Children: a Comprehensive Review. Maedica (Bucur). 2019 Sep;14(3):292-297.
- Poddar U. Helicobacter pylori: a perspective in low- and middle-income countries. Paediatr Int Child Health. 2019 Feb;39(1):13-17.
The authors should concentrate on the key elements that set their review apart from others that have been written on the same subject.
Additionally, I recommend English poolishing.Author Response
Response to Reviewer 1:
As per the reviewers’ recommendations, the manuscript has been substantively modified in accordance with the suggested changes. Of note, the major modification is that the abstract has been shortened. Our point-to-point responses to the reviewers are as follows.
1. This review aimed to demonstrate that H. pylori infection, iron insufficiency, and iron deficiency anemia are biologically likely connections. This assessment also tried to educate readers on H. pylori infection management, iron deficiency anemia, and the significance of appropriate clinical follow-up for these individuals. Although the review is well-written and interesting, I believe the abstract might be tighter and should concentrate on the most crucial information.
As indicated, the abstract has been shortened, made more concise and compressed, focusing on the most critically important (i.e. crucial) information.
2. There are several reviews on this topic such as: The authors should concentrate on the key elements that set their review apart from others that have been written on the same subject.
We understand the reviewer’s opinion and appreciate the suggestions on how to strengthen and make our review relevant with respect to what is currently in the literature. We believe that the present review addresses the most up to date clinical and basic evidence comprehensively which thereby provides substantiation and support for the biological basis and pathophysiologic mechanisms, in particular cause-and-effect, not just association, underlying H. pylori infection and iron deficiency and IDA. Further, we provide in our review the evidence which both clarifies and supports the causal association of H. pylori with IDA and its pathogenesis particularly in pediatric population, even if there might be overlaps between former review articles and ours.
3. Additionally, I recommend English polishing.
The manuscript has been re-reviewed again by the second author who is based in the U.S.A., Dr. Gold. The language of the manuscript now reflects those revision.
Reviewer 2 Report
This was a very well-done review study and the data reported is succinct and well-presented. But still had some scientific issue need further recognized.
1. In Introduction section, the H. pylori infection might relate to Nodular gastritis especially in adolescent or young adult patients. About 30-100% of children with nodular gastritis,90% with duodenal ulcer, and 25% with gastric ulcer disease are known to be infected with H. pylori. Authors might add this view point.
2. In line 339, authors had mentation the iron demand is important in H. pylori-associated IDA in adolescent female athletes. For adolescent female, they also need consider the menorrhagia and authors might consider add this in discussion.
Author Response
Response to Reviewer 2:
As per the reviewers’ recommendations, the manuscript has been substantively modified in accordance with the suggested changes. Of note, the major modification is that the abstract has been shortened. Our point-to-point responses to the reviewers are as follows.
1. In Introduction section, the H. pylori infection might relate to Nodular gastritis especially in adolescent or young adult patients. About 30-100% of children with nodular gastritis,90% with duodenal ulcer, and 25% with gastric ulcer disease are known to be infected with H. pylori. Authors might add this view point.
The authors thank Reviewer 2 for the comments which will assuredly strength our manuscript. In accordance with Reviewer’s comments in the introduction, we mentioned “In the pediatric population, H. pylori is also associated with the development of gastritis, peptic ulcer disease (duodenal more than gastric ulcers), in rare cases MALT lymphoma …” Since this review article focuses upon H. pylori and IDA, we simply described associations between H.pylori and the other disease.
2. In line 339, authors had mentation the iron demand is important in H. pylori-associated IDA in adolescent female athletes. For adolescent female, they also need consider the menorrhagia and authors might consider add this in discussion.
Like the author’s suggestion, menorrhagia is an important factor for the development of IDA in female. In Choe’s article, a possible association with H. pylori and IDA was statistically analyzed between the female athletes and the control group (female).
Reviewer 3 Report
General comments
This article is a good review covering basic science (mechanisms) and clinical evidence supporting a relationship between H. pylori infection and ID/IDA.
Although it is a general review (not systematic), it provides an integrated view especially of the mechanisms that could explain the association. The clinical evidence, on the other hand, appears weak in depth and analysis compared to the more basic or mechanistic sections.
Major issues
Section 2.2 Clinical evidence. It presents a series of information, somewhat disordered, mixing original works with meta-analysis. It also combines descriptive association studies with eradication studies. This section requires an important reorganization and separation or organization of the evidence:
- By types of studies (cross-sectional association, interventional, other).
- By age (adults vs. children)
- By geographic origin of the population studied).
- Then the meta-analyses
At least in Hudak's meta-analysis (ref 22) one has to be careful, as there was Significant heterogeneity among studies, as well as evidence of publication bias (Hudak 2016). And when analyzing separately ID from IDA, it is possible to observe differences as well:
ID: Separate meta-analyses of pediatric and adult studies that adjusted for confounders showed similar results (OR 1.43, 95% CI 1.15-1.78) and (OR 1.37, 95% CI 1.01-1.86), respectively, while such association was nonsignificant (OR 1.21, 95% CI 0.94-1.56; Pv=.13) in a meta-analysis of unadjusted results.
IDA: Pooled analysis of all 23 studies that addressed anemia as an outcome resulted in a pooled OR 1.15 (95% CI 1.00-1.32), Pv=.05. Significant heterogeneity was evident among the studies (Pv=.01). BUT, limiting the analysis to studies that adjusted for confounders in children, but not adults, showed a significantly increased likelihood of anemia by 26%: OR 1.26 (95% CI 1.02-1.55), with significant heterogeneity among the studies (Pv=.02).
I suggest to re view 2 additional MORE RECENT clinical evidence examining the relationship between iron deficiency/IDA and H pylori.
1. IDA was assessed in 2391 children from Turkey, in a cross-sectional study in children referred for UGI endoscopy. 699 children had anemia, 549 had low ferritin and no anemia and 1143 were control children. HPI (24% vs 17.6%, p=0.000) and Celiac disease (6% vs 2.2%; p=0.04) were more common in low ferritin levels without anemia compared to NO anemia (Bahardir 2021).
2. In a reduced group of 60 infected children from Israel, haemoglobin, and ferritin levels were compared prior and 6-9 months’ post-successful HP eradication (35% with IDA and 65% with ID). Iron normalized in 60% of patients with ID, without iron supplementation. Significant improvements were observed in Hgb (12.3 g/dL to 13.0 g/dL) and ferritin (6.3 μg/L to 15.1 μg/L) (p < 0.001) (Tanous 2021).
Closing paragraph (Perspectives). I believe that mentioning references 107 and 108 related to the topic of H pylori infection and growth failure/short stature is confusing and unhelpful. This topic deserves a separate and in-depth review since the literature has many confounding variables and H pylori infection is a marker of low economic status and it is difficult to establish causal relationships between short stature and H pylori infection.
Minor issues
Line 56: it seems to be a typo.... "and such as"....should be deleted.
Although this reviewer does not have English as a primary language, the text presents some grammatical difficulties in multiple paragraphs, especially sections 1 and 2, e.g., line 100.
If section 3.4 is RECENT pathogenic evidence, then the previous sections (3.1, 3.2, 3.3) should read OLD or CLASSIC or something similar.
Line 402....the comments regarding Maastrich V must be updated since it is already available to Consensus VI (2022).
Author Response
Response to Reviewer 3:
As per the reviewers’ recommendations, the manuscript has been substantively modified in accordance with the suggested changes. Of note, the major modification is that the abstract has been shortened. Our point-to-point responses to the reviewers are as follows.
Major issues
1. Section 2.2 Clinical evidence. It presents a series of information, somewhat disordered, mixing original works with meta-analysis. It also combines descriptive association studies with eradication studies. This section requires an important reorganization and separation or organization of the evidence:
Section 2.2 Clinical evidence. Thank you for your nice suggestions. As suggested, the paragraph has been reorganized in order to substantiate not just the biological plausibility of the epidemiological observations but provide evidence for cause-and-clinical effect between H. pylori infection and IDA.
2. At least in Hudak's meta-analysis (ref 22) one has to be careful, as there was Significant heterogeneity among studies, as well as evidence of publication bias (Hudak 2016). And when analyzing separately ID from IDA, it is possible to observe differences as well:
The sentence “H. pylori infection is also associated with increased likelihood of depleted iron stores [22]” was omitted. We appreciate the Reviewer’s insight and have offered the commentary on the limitations that are pointed out by this Reviewer with respect to the heterogeneity of the populations in Hudak meta-analysis.
3. I suggest to re view 2 additional MORE RECENT clinical evidence examining the relationship between iron deficiency/IDA and H pylori.
We appreciate Reviewer 3’s recommendations. As suggested, a new reference 25 (Tanous O, et al Acta Paediatr 2022) has been added and the previous ref. 25 was omitted.
4. Closing paragraph (Perspectives). I believe that mentioning references 107 and 108 related to the topic of H pylori infection and growth failure/short stature is confusing and unhelpful. This topic deserves a separate and in-depth review since the literature has many confounding variables and H pylori infection is a marker of low economic status and it is difficult to establish causal relationships between short stature and H pylori infection.
We agree with the Reviewer 3’s point that the literature has many confounding variables regarding the epidemiological association between H pylori infection and growth, and that other factors may be in fact more important. We also agree with this reviewer that H. pylori infection is also likely a surrogate marker for low economic status. The phrase “the literature has many confounding variables …” has been added. We referred to a possible association between H. pylori and malnutrition. It is our assertion that we hope further discussion and investigation into this important issue particularly in developing countries. To serve this purpose, we mentioned in the last sentence “Clearly further well-designed, ….”
Minor issues
1. Line 56: it seems to be a typo.... "and such as"....should be deleted.
As suggested, “and such as” was omitted.
2. If section 3.4 is RECENT pathogenic evidence, then the previous sections (3.1, 3.2, 3.3) should read OLD or CLASSIC or something similar.
In accordance with the reviewer’s suggestions, we used “recent” to intend “a hypothesis proposed more recently.”
3. Line 402....the comments regarding Maastrich V must be updated since it is already available to Consensus VI (2022).
Thank you for the reviewer’s comment. As suggested, Maastricht V/Florence consensus report was exchanged for Maastricht VI/Florence version 2022 (Ref. 104).
Round 2
Reviewer 1 Report
This review aims to provide evidence that supports a clear biological link between H. pylori infection and iron deficiency, as well as iron deficiency anemia.
The authors have answered to reviewers' comments in an appropriate way, and now it deserves to be published.